# Rare Phytocannabinoids Exert Anti-Inflammatory Effects on Human Keratinocytes via the Endocannabinoid System and MAPK Signaling Pathway

**DOI:** 10.3390/ijms24032721

**Published:** 2023-02-01

**Authors:** Daniel Tortolani, Camilla Di Meo, Sara Standoli, Francesca Ciaramellano, Salam Kadhim, Eric Hsu, Cinzia Rapino, Mauro Maccarrone

**Affiliations:** 1Department of Veterinary Medicine, University of Teramo, 64100 Teramo, Italy; 2European Center for Brain Research (CERC), Santa Lucia Foundation IRCCS, 00143 Rome, Italy; 3Department of Bioscience and Technology for Food Agriculture and Environment, University of Teramo, 64100 Teramo, Italy; 4InMed Pharmaceuticals Inc., Vancouver, BC V6C 1B4, Canada; 5Department of Biotechnological and Applied Clinical Sciences, University of L’Aquila, 67100 L’Aquila, Italy

**Keywords:** endocannabinoids, inflammation, keratinocytes, phytocannabinoids, signal transduction

## Abstract

Increasing evidence supports the therapeutic potential of rare cannabis-derived phytocannabinoids (pCBs) in skin disorders such as atopic dermatitis, psoriasis, pruritus, and acne. However, the molecular mechanisms of the biological action of these pCBs remain poorly investigated. In this study, an experimental model of inflamed human keratinocytes (HaCaT cells) was set up by using lipopolysaccharide (LPS) in order to investigate the anti-inflammatory effects of the rare pCBs cannabigerol (CBG), cannabichromene (CBC), Δ^9^-tetrahydrocannabivarin (THCV) and cannabigerolic acid (CBGA). To this aim, pro-inflammatory interleukins (IL)-1β, IL-8, IL-12, IL-31, tumor necrosis factor (TNF-β) and anti-inflammatory IL-10 levels were measured through ELISA quantification. In addition, IL-12 and IL-31 levels were measured after treatment of HaCaT cells with THCV and CBGA in the presence of selected modulators of endocannabinoid (eCB) signaling. In the latter cells, the activation of 17 distinct proteins along the mitogen-activated protein kinase (MAPK) pathway was also investigated via Human Phosphorylation Array. Our results demonstrate that rare pCBs significantly blocked inflammation by reducing the release of all pro-inflammatory ILs tested, except for TNF-β. Moreover, the reduction of IL-31 expression by THCV and CBGA was significantly reverted by blocking the eCB-binding TRPV1 receptor and by inhibiting the eCB-hydrolase MAGL. Remarkably, THCV and CBGA modulated the expression of the phosphorylated forms (and hence of the activity) of the MAPK-related proteins GSK3β, MEK1, MKK6 and CREB also by engaging eCB hydrolases MAGL and FAAH. Taken together, the ability of rare pCBs to exert an anti-inflammatory effect in human keratinocytes through modifications of eCB and MAPK signaling opens new perspectives for the treatment of inflammation-related skin pathologies.

## 1. Introduction

In the last few years, the use of botanical ingredients in dermatology has been continuously increasing; as of now, studies have found different botanical substances both effective and safe, although more studies and clinical trials are needed [1]. A plant that has a long history of medical as well as recreational purposes is cannabis. Recently, the interest in its medical aspects has increased, and the legalization of cannabis in a growing number of countries gave rise to the possibility of using cannabinoids in healthcare and skincare formulation [2], although the effects of the different components, including less abundant (rare) cannabinoids, are not yet clear. Cannabinoids can be divided into three categories: phytocannabinoids (pCBs) produced by plants, endocannabinoids (eCBs) biosynthesized in the human body, and synthetic cannabinoids generated through chemical processes [3]. In particular, pCBs are terpenoids consisting of 21 or 22 carbon atoms that usually contain a propyl or pentyl side chain [4]. In cannabis, which is an annual, pollinated, flowering plant from the *Cannabaceae* family, pCBs are synthesized and stored in the glandular trichomes found at the highest density in the female flowers of the plant [4]. Up to 120 pCBs have been identified so far, among which the best studied are Δ^9^-tetrahydrocannabinol (THC), cannabidiol (CBD) and cannabigerol (CBG) [5]. Recent studies have focused on rare pCBs and their potential therapeutic exploitation [6]. In particular, the analgesic and anti-inflammatory properties of cannabis extracts were known for decades [7], and pCBs have attracted interest for the treatment of different dermatologic disorders [2].

Rare pCBs modulate the endogenous eCB system of human skin [8], which is a signaling network consisting of eCBs, their metabolic enzymes and receptor targets [9,10], with apparent potential for the treatment of skin diseases [11]. The two main eCBs, *N*-arachidonoylethanolamine (anandamide; AEA) and 2-arachidonoylglycerol (2-AG), can bind to and activate G protein-coupled type 1 (CB_1_) and type 2 (CB_2_) cannabinoid receptors as well as to non-cannabinoid targets such as the transient receptor potential vanilloid type 1 (TRPV1) channel, the nuclear peroxisome proliferator-activated receptors (PPARs) and the orphan G protein-coupled receptor 55 (GPR55) [12]. Although there are several metabolic pathways for eCBs [13], AEA is mostly biosynthesized by the *N*-acylphosphatidylethanolamine-specific phospholipase D (NAPE-PLD) and is principally degraded by the fatty acid amide hydrolase (FAAH) [14]. Instead, the biosynthesis of 2-AG can occur through multiple reactions that generate diacylglycerol (DAG), which is converted by diacylglycerol lipases α and β (DAGLα/β) into 2-AG and then hydrolyzed into arachidonic acid and glycerol mainly by the monoacylglycerol lipase (MAGL) [15].

Of note, all of the major ECS elements previously cited can be found in the skin and are synthesized and expressed by different tissues and cell types—such as keratinocytes, melanocytes, dermal fibroblasts, adipocytes, immune cells, nerve fibers and skin appendages [16,17]—overall supporting the view that they are actively involved in the complex pathophysiology of the skin [18].

The skin represents the outermost barrier of the body, providing a physical shield and the first defense against external pathogens [19]. As the most dominant cell type in the skin, keratinocytes actively maintain homeostasis in the epidermis and restore it after injuries, acting as sentinels with crucial immune functions during inflammatory responses [20]. Indeed, keratinocytes express a broad range of pattern-recognition receptors (PRRs), such as the Toll-like receptors (TLRs), able to recognize specific molecules known as pathogen-associated molecular patterns (PAMPs) such as the bacterial lipopolysaccharides (LPS) that are frequently found in foreign pathogens [21]. PRRs activation leads to the production of pro-inflammatory cytokines and chemokines with subsequent immune cell recruitment, which is implicated in the process of wound healing and re-epithelialization [22]. In this context, eCB signaling is engaged in the regulation of skin cell proliferation, survival and differentiation, thus playing a crucial role in proper cutaneous homeostasis [9,10]. Furthermore, eCBs exert a protective role against allergic inflammation of the skin [9] and can be involved in both acute and chronic inflammatory diseases [23].

Against this background, we aimed here to explore the possible effects of four rare, non-psychotropic pCBs, namely CBG, CBC, Δ^9^-tetrahydrocannabivarin (THCV) and cannabigerolic acid (CBGA), on acute inflammation induced in vitro by LPS. To this end, immortalized human keratinocytes (HaCaT cells) were used because they express a complete and functional eCB system [24] that is modulated by pCBs, as previously demonstrated [8]. In particular, rare pCBs increased CB_1/2_ binding, TRPV1 channel stimulation and FAAH and MAGL catalytic activity, and they differently modulated gene and protein expression of distinct ECS elements as well as the content of eCB(-like) compounds [8]. In the present study, the effects of CBG, CBC, THCV and CBGA were ascertained by observing the expression of specific pro-inflammatory cytokines, and the effect of THCV and CBGA on IL-12 and IL-31 was further investigated in the presence of selected modulators of eCB signaling. Finally, the effect of THCV and CBGA was ascertained on 17 different mitogen-activated protein kinases (MAPKs), which were also in the presence of selected antagonists and inhibitors of distinct eCB receptors and enzymes.

## 2. Results

### 2.1. Set Up of an In Vitro Model of Inflamed Keratinocytes (HaCaT Cells)

First, an experimental model of inflamed human keratinocytes (HaCaT cells) was set up by using different amounts of LPS (1.0, 5.0, 10.0 μg/mL) for 24 h and 48 h. Cell viability was checked by using Trypan blue dye exclusion, demonstrating that LPS was not cytotoxic at any dose used (data not shown). As an inflammatory read-out, the gene expression of cyclooxygenase-2 (COX-2), an enzyme known to induce pro-inflammatory responses to LPS [25], was evaluated by using quantitative real-time PCR (RT-qPCR). In particular, COX-2 expression was significantly increased following: (i) 24 h LPS treatment at doses of 1.0 μg/mL (*p* < 0.001 vs. CTRL), 5.0 μg/mL and 10.0 μg/mL (both *p* < 0.0001 vs. CTRL), and (ii) 48 h LPS treatment at the two doses of 5.0 μg/mL and 10.0 μg/mL (*p* < 0.05 and *p* < 0.0001 vs. CTRL, respectively) (Figure 1A,B). In addition, the release of the two pro-inflammatory cytokines interleukin (IL)-8 and IL-12 into the cell culture medium was evaluated by using ELISA tests. IL-8 protein content was found to be significantly increased in a dose-dependent manner at both 24 h and 48 h in HaCaT cells treated with different concentrations of LPS (1.0, 5.0 and 10.0 μg/mL) (*p* < 0.05, *p* < 0.001 and *p* < 0.0001 vs. CTRL at 1.0, 5.0, 10.0 μg/mL of LPS, respectively, for 24 h; *p* < 0.001, *p* < 0.0001 vs. CTRL at 1.0 and 5.0 μg/mL, 10.0 μg/mL of LPS, respectively, for 48 h), and IL-12 content was also higher (*p* < 0.05 and *p* < 0.001 vs. CTRL at 5.0 and 10.0 μg/mL of LPS, respectively, for 24 h; *p* < 0.001 and *p* < 0.0001 at 5.0 and 10.0 μg/mL of LPS, respectively, for 48 h) except upon treatment with LPS 1.0 μg/mL, which did not induce any modification (Figure 1E,F). Based on these data, the LPS concentration of 5.0 μg/mL was chosen as the minimum effective dose able to induce a significant increase in inflammatory markers after 24 and 48 h of treatment.

### 2.2. Effects of Rare pCBs on Cytokine Expression

To evaluate the potential anti-inflammatory effect of CBG, CBC, THCV and CBGA on HaCaT cells stimulated with LPS at the selected dose of 5.0 μg/mL for 24 h and 48 h, the protein content of different pro-inflammatory interleukins (IL-1β, IL-8, IL-12 and IL-31), tumor necrosis factor (TNF-β) and anti-inflammatory IL-10 was assessed via ELISA. Based on our previous studies on the same cell line, pCBs were used at the following non-cytotoxic concentrations: 6.0 µM for CBG, 4.0 µM for CBC, 9.3 µM for THCV and 13.0 µM for CBGA [8]. In addition, exposure of HaCaT cells to 10 μM hydrocortisone (HC), a known anti-inflammatory agent [26], was included as a positive control. All interleukin levels released in the culture medium of HaCaT cells, treated or not treated with pCBs, were interpolated from the corresponding standard curves (shown in Appendix A). Our results show that CBGA significantly decreased the release of IL-1β after 24 h of treatment with LPS (*p* < 0.01 vs. LPS) (Figure 2A), whereas CBG and CBC significantly increased the levels of IL-1β after 48 h treatment (*p* < 0.01 and *p* < 0.05 vs. LPS) (Figure 2B). Moreover, all pCBs significantly reduced the release of the pro-inflammatory IL-8 from HaCaT cells treated with LPS for 24 h (CBG *p* < 0.0001; CBC, THCV and CBGA *p* < 0.01 vs. LPS) (Figure 2C). However, such a reduction appeared to be transient, and it was not observed after 48 h of treatment (Figure 2D). Upon 48 h of LPS exposure, all pCBs were able to reduce the expression of the pro-inflammatory IL-12 (CBG and CBC *p* < 0.001; THCV and CBGA *p* < 0.0001 vs. LPS) (Figure 2F). Moreover, THCV and CBGA led to a downregulation of the pro-inflammatory IL-31 after 24 h treatment with LPS (THCV *p* < 0.05; CBGA *p* < 0.01 vs. LPS) (Figure 2G), and conversely, CBGA showed a trend towards an increase in the release of IL-31 after 48 h (Figure 2H). Finally, no significant changes were observed in the expression of IL-10 (Figure 2J,K) and TNF-β (Figure 2M,N) at either time of exposure, suggesting a non-inducible expression of these cytokines. A summary of the effects induced by each pCB on cytokine release is shown in Table 1.

### 2.3. Role of the eCB System in Cytokine Release Mediated by THCV and CBGA

Based on the results summarized in Table 1, we sought to better investigate the mechanism of action of THCV and CBGA, as they are the pCBs with the most marked effects on IL-12 expression and the only ones that determined significant modification of IL-31 expression. The goal was to ascertain whether the anti-inflammatory effect of THCV and CBGA occurred through modulation of eCB system elements such as metabolic enzymes and TRPV1 receptors, as already observed in unstimulated HaCaT cells [8]. To this aim, protein levels of IL-31 and IL-12 were measured at 24 h and 48 h, respectively, upon exposure of HaCaT cells to LPS alone or in the presence of THCV and CBGA as well as of selected eCB receptor antagonists or eCB enzyme inhibitors. In particular, a selected TRPV1 antagonist (CPZ) was used for both pCBs, selected inhibitors of MAGL (JZL184) and DAGLα/β (LEI-106) were used for THCV, and selected inhibitors of FAAH (URB597) and NAPE-PLD (ARN19874) were used for CBGA at doses reported in the literature (for CPZ, [27]; for JZL184 and LEI-106, [28]; for URB597, [29] and for ARN19874, [30]). These eCB system modulators were selected on the basis of a previous study, wherein TRPV1 activation by both pCBs and modulation of 2-AG and AEA metabolism by THCV and CBGA, respectively, were shown in HaCaT cells [8].

Regarding IL-31, a significant decrease compared to LPS alone was observed after treatment with both pCBs for 24 h (*p* < 0.01 vs. LPS for THCV; *p* < 0.0001 vs. LPS for CBGA) (Figure 3A). Remarkably, the latter effect was reverted by CPZ (*p* < 0.01 vs. THCV; *p* < 0.0001 vs. CBGA), suggesting an involvement of the TRPV1 channel in the anti-inflammatory action of both pCBs. In addition, the selected inhibitor of MAGL (JZL184) was able to significantly revert the action of THCV (*p* < 0.001 vs. THCV). Regarding IL-12, the results showed significant reduction after treatment with THCV and CBGA for 48 h, with respect to LPS (*p* < 0.01 vs. LPS), yet none of the antagonists or inhibitors used in combination with each pCB affected IL-12 levels (Figure 3B).

### 2.4. Effects of THCV and CBGA on the eCB-Dependent MAPK Signaling Pathway

The eCB-dependent MAPK signaling pathway in HaCaT cells treated with LPS, either alone or in combination with THCV and CBGA, was investigated in the presence of inhibitors of selected eCB enzymes and an antagonist of the TRPV1 receptor. Much like the experiments on cytokines, the selected TRPV1 antagonist CPZ for both pCBs, selected inhibitors of MAGL (JZL184) and DAGLα/β (LEI-106) for THCV and selected inhibitors of FAAH (URB597) and NAPE-PLD (ARN19874) for CBGA were used. Then, the MAPK signaling pathway was analyzed through a Human Phosphorylation Array able to detect the expression of the following 17 proteins that are recognized as important players along the MAPK transduction route [31]: serine/threonine kinase 1 (AKT); cyclic adenosine monophosphate (cAMP); response element-binding protein (CREB); glycogen synthase kinase 3α (GSK3α) and 3β (GSK3β); c-Jun N-terminal kinase (JNK); extracellular signal-regulated kinase (ERK1); mitogen-activated protein kinase (MEK1); mitogen-activated protein kinase 3 (MKK3) and 6 (MKK6); mitogen- and stress-activated protein kinase 2 (MSK2); heat shock protein 27 (HSP27); mammalian target of rapamycin (mTor); p38 mitogen-activated protein kinase (p38); tumor suppressor protein (p53); p70 ribosomal S6 kinase (P70S6k); ribosomal S6 kinase 1 (RSK1) and 2 (RSK2) (for details, see Appendix A). It should be noted that all proteins were analyzed in their active phosphorylated forms except for GSK3β, which is inactive when phosphorylated [32]. Incidentally, GSK3β is a serine/threonine protein kinase known to act as a key downstream regulatory switch for numerous signaling pathways [33]. Overall, the heat map analysis showed low expression of all phosphorylated proteins of the MAPK pathway in HaCaT cells treated with LPS, either alone or in the presence of pCBs and selected antagonists and inhibitors to eCB system elements. The only exception was GSK3β, which was instead highly expressed under all experimental conditions (Figure 4).

At any rate, AKT, GSK3β and mTor expression levels were significantly reduced (*p* < 0.05 vs. CTRL), whereas MEK1 and MKK6 expression levels were significantly increased (*p* < 0.05 vs. CTRL) in LPS-treated keratinocytes compared to controls (Figure 5, and Appendix A). In these cells, CBGA was able to significantly increase AKT (*p* < 0.01 vs. LPS), GSK3β (*p* < 0.05 vs. LPS) and mTor (*p* < 0.001 vs. LPS) expression when the inflamed HaCaT cells were treated with ARN19874 (for AKT, Figure 5A), URB597 and CPZ (for GSK3β, Figure 5B) and URB597 (for mTor, Figure 5C). On the other hand, MEK1 expression was significantly reduced in inflamed keratinocytes exposed to CBGA alone (*p* < 0.01 vs. LPS) and in the presence of CPZ (*p* < 0.001 vs. LPS) as well as in cells treated with THCV alone (*p* < 0.01 vs. LPS) and in the presence of JZL184 (*p* < 0.01 vs. LPS) and LEI-106 (*p* < 0.05 vs. LPS) (Figure 5D). Similarly, MKK6 expression was remarkably decreased by both CBGA and THCV under all experimental conditions compared to LPS (*p* < 0.05; *p* < 0.01; *p* < 0.001 vs. LPS) (Figure 5E). Finally, THCV was able to significantly increase mTor expression in cells treated with CPZ and JZL184 (*p* < 0.05 vs. LPS) (Figure 5C).

Incidentally, the expression of the MAPK signaling proteins CREB, RSK1 and RSK2 (Figure 6) was not significantly modified by LPS compared to controls except for those inflamed cells exposed to treatments with both CBGA and THCV in the presence of eCB system modulators. In particular, CREB expression was significantly reduced in inflamed keratinocytes exposed to THCV and CBGA alone (*p* < 0.01 vs. LPS) as well as in the presence of LEI-106 for THCV (*p* < 0.01 vs. LPS) and of CPZ for CBGA (*p* < 0.01 vs. LPS) (Figure 6A). Conversely, JZL184 and URB597 were able to significantly revert CREB expression when cells were treated with THCV (*p* < 0.01 vs. THCV) and CBGA (*p* < 0.001 vs. CBGA), respectively (Figure 6A). Regarding RSK1 (Figure 6B) and RSK2 (Figure 6C), CBGA significantly increased both proteins but only in the presence of URB597 (*p* < 0.05 for RSK1 and *p* < 0.001 for RSK2 vs. CBGA).

Overall, the effects of the selected elements of the eCB system on the expression of the MAPK signaling proteins are schematically reported in Table 2.

## 3. Discussion

Cannabis and its extracts (pCBs, terpenes, phenols, etc.) have historically shown anti-inflammatory properties [34]; during the last few years, more studies have addressed the potential application of pCBs on several pathologies, including skin diseases [2]. In this context, we previously demonstrated the ability of some rare pCBs (CBG, CBC, THCV and CBGA) to modulate the main eCB system elements in unstimulated human keratinocytes (HaCaT cells) [8,35]. Against this background, the anti-inflammatory effect of CBG, CBC, THCV and CBGA in an in vitro LPS-induced model of inflammation was shown here by demonstrating the reduction of all pro-inflammatory cytokines tested, except for TNF-β. Of interest, among all ILs tested, IL-8, IL-12 and IL-31 seem to be involved in several inflammatory skin conditions, such as psoriasis, atopic dermatitis and itch, as well as in innate and adaptive immunity [36,37,38,39,40]. Here, the inhibition by pCBs of the release of these interleukins appears to be time-dependent; in particular, IL-8 and IL-31 seem to be affected at an early time point (24 h), whereas IL-12 seems to be altered at a later stage (48 h), suggesting a delayed anti-inflammatory role for the latter. Incidentally, IL-12 was reported to initiate a tissue-protective response in inflamed keratinocytes and to counter-regulate the psoriatic transcriptional signature in both murine and human keratinocytes [41].

Particular attention is called to IL-31, which plays a prominent role in the itch sensation in keratinocytes via induction of leukotriene B_4_, and it is implicated in the pathogenesis of psoriasis [39]. In the in vitro inflamed HaCaT cells, it can be concluded that IL-31 expression was markedly reduced by THCV and CBGA through a TRPV1-dependent mechanism because the specific antagonist of this receptor (CPZ) reverted the effect of both pCBs. Of note, TRPV1 was found to also be activated by THCV in non-inflamed HaCaT cells [8]. Moreover, THCV was considered a potent anti-acne agent, as it revealed significant lipostatic activity, suppressed sebocytes proliferation and abrogated LPS-induced pro-inflammatory responses [42]. Instead, a specific role of CBGA in inflammatory skin diseases has never been attested. Regarding the TRPV1 channel, its overexpression was observed in the skin of psoriatic patients with severe itch [43] and in the mouse model of imiquimod (IMQ)-induced psoriasiform dermatitis (PsD) [44]. In addition, TRPV1 was found to be a critical mediator of persistent itch in a mouse model of squaric acid dibutylester-induced contact dermatitis (SADBE) [45]. Here, our results also show the ability of MAGL inhibition to counteract the anti-inflammatory action of THCV on IL-31 levels; thus, the resulting increased level of 2-AG can play a key role in skin inflammation. In line with this, the systemic and spinal administration of the MAGL inhibitor JZL184 produced an antipruritic effect in mice [46], and a notable increase of 2-AG levels in skin lesions was observed in mice affected by mite antigen-induced dermatitis compared to controls [47].

In addition, of note seems to be the unprecedented observation that the expression of the phosphorylated forms of several MAPK-related proteins was modified by THCV and CBGA by engaging specific eCB system elements. In particular, the phosphorylated (inactive) form of GSK3β at serine 9 (*p*-Ser9) appeared to be largely modulated and showed the highest expression values among all of the MAPK-related proteins tested. It should be noted that GSK3β is a potent driver of inflammation in its non-phosphorylated (active) form [48], and indeed, the phosphorylation of serine 9 is the major regulatory checkpoint for its activity [49]. Regarding the CBGA effect, it is conceivable that this rare pCB participates synergistically with AEA—increased upon FAAH inhibition—to inactivate the inflammatory action of GSK3β by promoting its Ser9 phosphorylation. Alternatively, CBGA could exert this effect on GSK3β synergistically with TRPV1 inactivation. Furthermore, THCV and CBGA could also play an anti-inflammatory role by dysregulating the MAPK-related signaling proteins CREB, MEK1 and MKK6 as well as by engaging 2-AG- and AEA-metabolizing enzymes (MAGL and FAAH, respectively). Of note, the inhibition of MAGL restored the anti-inflammatory effect THCV exerted on CREB, further supporting the IL-31 data on the importance of endogenous 2-AG levels in skin inflammation. Instead, the anti-inflammatory effect of CBGA on CREB was reverted via FAAH inhibition, suggesting in this case a prominent role for the endogenous AEA level. In this context, CREB appears to be involved in several processes in immune as well as non-immune cells, promoting both pro- and anti-inflammatory responses [50]. For instance, a study showed CREB activation upon treatment of mouse skin with a proinflammatory agent [51], and increased CREB activity was demonstrated in lesioned psoriatic epidermis and psoriatic keratinocytes [52,53]. In addition, MEK1 signaling was found to be abnormally activated in inflammatory skin diseases and cancer, and so was MKK6, whose dysregulation can lead to the pathogenesis of a range of inflammatory diseases [54]. Incidentally, it should be stressed that the use of signal transduction arrays is only the first step to investigate possible changes in wide sets of proteins, aimed at identifying specific targets that should be further investigated in independent studies via complementary analytical methods [55].

In summary, our present results have identified THCV and CBGA as anti-inflammatory agents that reduce pro-inflammatory cytokines such as IL-31 by interacting with the eCBs system as well as the MAPK signaling pathway. Overall, GSK3β, MEK1, MKK6 and CREB appear to be unprecedented targets of these rare pCBs and seem to deserve further analysis in additional independent studies.

## 4. Materials and Methods

### 4.1. Materials

The phytocannabinoids (CBG, CBC, THCV and CBGA) were purchased from Cerilliant Corporation (Sigma-Aldrich Company, St. Louis, MO, USA). Cell culture reagents, including high-glucose Dulbecco’s Modified Eagle Medium (DMEM-HG) and fetal bovine serum (FBS), were purchased from Corning Incorporated (Corning, NY, USA). Antibiotics (Pen/Strep), phosphate buffer saline (D-PBS without calcium and magnesium) and trypsin (2.5%) were obtained from Gibco by Life Technologies (Thermo Fisher Scientific Company, Waltham, MA, USA), and EDTA (0.5 M) was obtained from Invitrogen (Thermo Fisher Scientific Company, Waltham, MA, USA). Lipopolysaccharides (LPS) from *Escherichia Coli* O111:B4 (suitable for cell culture) and Hydrocortisone (HC) were purchased from Sigma Aldrich (St. Louis, MO, USA). The antagonist CPZ and the inhibitor URB597 were obtained from Sigma Aldrich (St. Louis, MO, USA), whereas the inhibitors ARN19874, JZL184 and LEI-106 were obtained from Cayman Chemical (Ann Arbor, MI, USA). The kit for the enzyme-linked immunosorbent assays (ELISA) for each interleukin was purchased from R&D System DuoSet (Bio-Techne, Minneapolis, MN, USA), and the Human/Mouse MAPK Phosphorylation Array was obtained from RayBiotech (RayBiotech Inc., Peachtree Corners, GA, USA). Supplies for the quantitative real-time polymerase chain reaction (RT-qPCR), including the RevertAid H Minus First Strand cDNA Synthesis Kit and SensiFAST SYBR Lo-ROX Kit, were obtained from Thermo Fisher Scientific (Waltham, MA, USA) and Bioline (Meridian Bioscience Inc. Company, Cincinnati, OH, USA), respectively.

### 4.2. Cell Line and Treatments

Immortalized HaCaT cells from the original depositor (DKFZ, Heidelberg), which were of Caucasian skin type (phototype), were purchased from CLS-Cell Lines Service (code 300493). In all of the experiments, HaCaT cells were used at the 5th–6th passage, cultured at 37 °C in a humidified 5% CO_2_ atmosphere in high-glucose Dulbecco’s Modified Eagle Medium (DMEM-HG) and supplemented with a 10% fetal bovine serum (FBS) and 1% antibiotic-antimycotic (pen/strep) solution. For each treatment, after 24 h of cell growth, the cell culture medium was replaced with DMEM-HG supplemented with 1% FBS and 1% pen/strep (Starvation medium) and incubated for 24 h. On the third day, the cells were incubated with the following conditions, according to the treatment of interest: control (DMEM + 5% FBS + 1% pen/strep); LPS at 1.0, 5.0 and 10.0 μg/mL; positive control Hydrocortisone (HC) at 10.0 μM; pCBs at specific pre-calculated concentrations (CBG 6.0 μM, CBC 4.0 μM, THCV 9.3 μM and CBGA 13.0 μM) [8] and the same pCBs in combination with the following selected eCB system antagonists and inhibitors: CPZ at 5.0 μM [27]; JZL184 at 10.0 μM and LEI-106 at 10.0 μM [28]; URB597 at 1.0 μM [29] and ARN19874 at 33.7 μM [30]. After 24 and 48 h, cell pellets and supernatants (medium) from each condition were collected, centrifuged at 300× *g* for 5 min, and stored at −20 °C and at −80 °C, respectively, after centrifugation.

### 4.3. Quantitative Real-Time Polymerase Chain Reaction (RT-qPCR)

HaCaT cells were seeded at a density of 4 × 10^5^ per well, incubated overnight and then exposed to LPS treatment at the three concentrations of 1.0, 5.0 and 10.0 µg/mL for 24 h and 48 h. Afterwards, cell pellets were collected and stored at −20 °C. Total RNA was extracted from the cell pellets by using QIAzol Lysis Reagent (Qiagen, Hilden, Germany), according to manufacturer’s instructions, and after extraction, it was quantified with a NanoDrop™ 2000/2000c Spectrophotometer (Thermo Fisher Scientific Company, Waltham, MA, USA). Using the RevertAid H Minus First Strand cDNA Synthesis (Thermo Fisher Scientific Company, Waltham, MA, USA), 500 ng of isolated mRNA was reverse transcribed in a final volume of 20 µL, later diluted 1:3, following incubation in a thermocycler for 5 min at 65 °C, 5 min at 25 °C, 60 min at 42 °C and 5 min at 70 °C. RT-qPCR was performed using the SensiFAST SYBR Lo-ROX Kit (Bioline by Meridian Bioscience Inc., Cincinnati, OH, USA) in an Applied Biosystems 7500 Fast Real-Time PCR System (Life Technologies, Carlsbad, CA, USA). Each PCR reaction (10 µL final volume) was carried out with 1 µL of cDNA, 1 µL of primers Forward + Reverse (10 µM mix) and 5 µL of Sybr Green 1X. The qPCR reaction was performed with the following conditions: holding at 50 °C for 2 min and 95 °C for 10 min, cycling at 95 °C for 15 s and 60 °C for 30 s (for 40 cycles) and, finally, a dissociation curve (melting curve) was constructed in the range of 60 °C to 95 °C [56], as detailed in [8]. The relative expression of the different amplicons was calculated using the ΔΔCt method and then converted into the relative expression ratio (2^−ΔΔCt^) for statistical analysis [57]. All data were normalized to the endogenous reference genes β-actin and glyceraldehyde-3-phosphate dehydrogenase (GAPDH). Primers used for COX-2, β-actin and GAPDH were designed with Primer3 and ordered from Integrated DNA Technologies (IDT; Coralville, IA, USA), and their specific sequences are reported in Table 3.

### 4.4. Enzyme-Linked Immunosorbent Assay (ELISA)

HaCaT cells were seeded into 12-well plates at a density of 2 × 10^5^ per well, incubated overnight and then exposed to LPS at 5.0 µg/mL, to positive control Hydrocortisone (HC) at 10.0 μM and to selected pCBs at pre-calculated concentrations (CBG 6.0 μM, CBC 4.0 μM, THCV 9.3 μM and CBGA 13.0 μM) corresponding to half of the IC_50_ values [8] for 24 h and 48 h. In addition, HaCaT cells were exposed to THCV and CBGA in combination with selected eCB system antagonists and inhibitors (CPZ 5.0 μM for both pCBs, JZL184 10.0 μM for THCV, LEI-106 10.0 μM for THCV, URB597 1.0 μM for CBGA and ARN19874 33.7 μM for CBGA) for 24 h in the case of IL-31 and for 48 h in the case of IL-12. After each treatment, supernatants (medium) of each condition were collected, centrifuged at 4 °C at 300× *g* for 5 min and stored at −80 °C until the assay. Cytokine quantification was performed using the DuoSet ELISA kit (R&D Systems, Minneapolis, MN, USA) for IL-1β, IL-8, IL-10, IL-12 and IL-31, following the manufacturer’s instructions. Briefly, 96-well plates were coated overnight with the specific capture antibody for each interleukin and then blocked with 1% BSA. Standard proteins in duplicate and samples in triplicate were added and incubated for 2 h. The standard curve for each interleukin was generated by using serially diluted standard proteins at known concentrations as provided by the manufacturer, and the calibration curves obtained were used to interpolate sample values. Then, each well with standards and/or samples was incubated with the specific detection antibody for 2 h. The Streptavidin-HRP was added and incubated for 20 min, followed by the substrate solution for another 20 min, and then the reaction was stopped with the specific Stop Solution (1M H_2_SO_4_). As for TNF-β, quantification was performed using the Human TNF-beta Platinum ELISA 96 tests (Invitrogen, Waltham, MA, USA). Briefly, 96-well plates already coated with TNF-β capture antibody were incubated for 4 h with standard and samples along with HRP-conjugate. After this period, TMB Substrate Solution was added to all wells for 10 min of incubation and then stopped with the specific Stop Solution (1M H_3_PO_4_). Optical densities were determined using an Enspire microplate reader (Perkin Elmer, Waltham, MA, USA) at the specific wavelengths of 450 and 570 nm for IL-1β, IL-8, IL-10, IL-12 and IL-31 and of 450 nm and 620 nm for TNF-β.

### 4.5. MAPK Signaling Pathway Array

HaCaT cells were seeded into 100 mm plates at a density of 8 × 10^5^ per well, incubated overnight and then exposed to 24 h treatment with LPS at 5.0 µg/mL, positive control Hydrocortisone (HC) at 10.0 μM, pCBs THCV (9.3 μM) and CBGA (13.0 μM) [8], and the same pCBs in combination with selected eCB system antagonists and inhibitors (CPZ 5.0 μM for both pCBs, JZL184 10.0 μM for THCV, LEI-106 10.0 μM for THCV, URB597 1.0 μM for CBGA and ARN19874 33.7 μM for CBGA). This signaling pathway was analyzed using a C-Series Human/Mouse MAPK Phosphorylation Array, according to manufacturer recommendations (RayBiotech Inc., Peachtree Corners, GA, USA). Briefly, proteins were extracted from HaCaT cell pellets and quantified using the BCA method. Total protein in the amount of 500 micrograms was added into each nitrocellulose membrane well coated with the specific antibody and incubated overnight at 4 °C. The day after, each well was incubated with the detection antibody and Streptavidin-HRP for two hours at room temperature. Chemiluminescent readings were taken using a C-DiGit Blot Scanner (LI-COR Bioscience, Lincoln, NE, USA), and densitometry data were extracted using Image Studio™ software (LI-COR Bioscience, Lincoln, NE, USA). Readings were normalized to the positive loading controls, and the membrane background signal was subtracted. Appendix A shows the distribution of antibodies in the membrane of the MAPK array.

### 4.6. Statistical Analysis

Data were analyzed using the GraphPad Prism 8 program (GraphPad Software, La Jolla, CA, USA) and were reported as means ± SEM of three independent experiments. Statistical analysis was performed using the one-way analysis of variance (ANOVA) test followed by a Bonferroni post-hoc test. A level of *p* < 0.05 was considered statistically significant.

## 5. Conclusions

In conclusion, we propose that the in vitro (LPS-induced) model of inflamed HaCaT cells can be used by measuring distinct pro-inflammatory cytokines—such as IL-31—to establish the anti-inflammatory potential of selected pCBs—such as THCV and CBGA—and their ability to engage eCB-binding receptors and metabolic enzymes.

Of note, we show that THCV and CBGA can act synergistically with AEA and 2-AG metabolic enzymes (MAGL and FAAH, respectively) to activate distinct proteins along the anti-inflammatory MAPK signaling pathway. Overall, this proof of concept, which shows that in inflamed human keratinocytes, rare pCBs can indeed interact with specific eCB system elements, opens new perspectives for possible treatments of inflammation-related skin diseases. Incidentally, such interactions between pCBs and eCB system seems to hold therapeutic potential well beyond the skin, such as possible treatments reported for autism spectrum disorders [58] and cancer during the preparation of this manuscript [59].

## Figures and Tables

**Figure 1 ijms-24-02721-f001:**
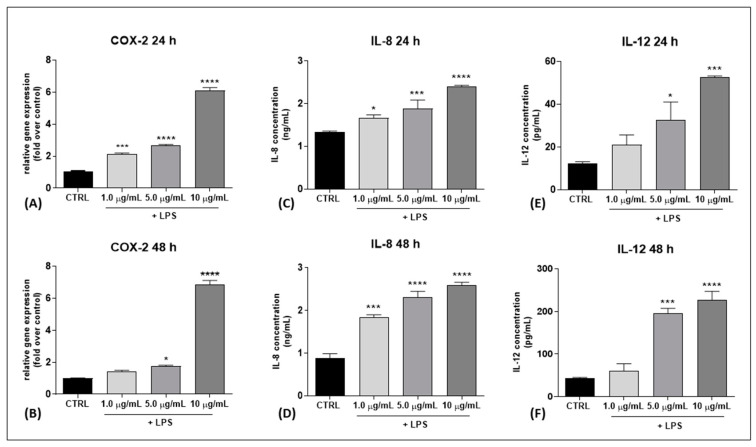
Gene expression of the inflammatory marker COX-2 and release of the pro-inflammatory cytokines IL-8 and IL-12 into the culture medium of HaCaT cells following LPS treatment (at 1.0, 5.0, 10.0 µg/mL) for 24 h and 48 h. (**A**,**B**) COX-2 gene expression at 24 h and 48 h, respectively. The values were expressed as 2(^−ΔΔCt^) and were normalized to β-actin and GAPDH as housekeeping genes. (**C**–**F**) Release of IL-8 (ng/mL) and IL-12 (pg/mL) at 24 h and 48 h, respectively. All of the data are presented as the means ± SEM of three independent experiments (*n* = 3). Statistical analysis was performed using a one-way ANOVA test followed by a Bonferroni post hoc test. (*****
*p* < 0.05; *******
*p* < 0.001; ********
*p* < 0.0001 vs. CTRL).

**Figure 2 ijms-24-02721-f002:**
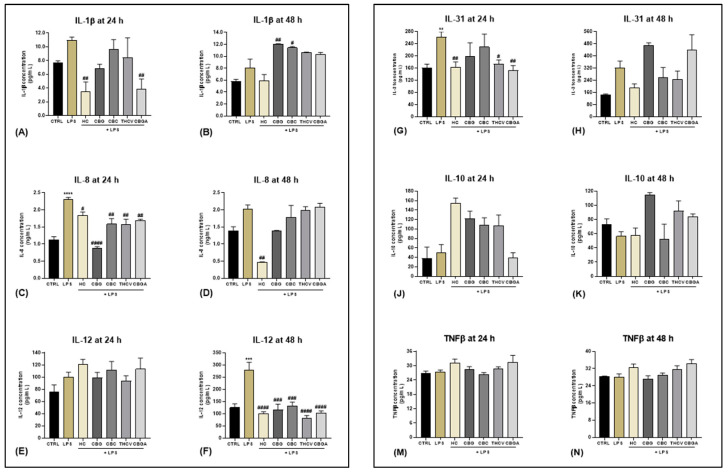
Release of cytokines into HaCaT cells culture medium. IL-1β expression (pg/mL) in HaCaT cells after (**A**) 24 h and (**B**) 48 h of treatment with LPS at 5.0 µg/mL either in the presence of HC at 10.0 µM or the following pCBs: CBG 6.0 µM, CBC 4.0 µM, THCV 9.3 µM and CBGA 13.0 µM; IL-8 expression (ng/mL) in HaCaT cells after (**C**) 24 h and (**D**) 48 h of treatment; IL-12 expression (pg/mL) in HaCaT cells after (**E**) 24 h and (**F**) 48 h of treatment; IL-31 expression (pg/mL) in HaCaT cells after (**G**) 24 h and (**H**) 48 h of treatment; IL-10 expression (pg/mL) in HaCaT cells after (**J**) 24 h and (**K**) 48 h of treatment; TNF-β expression (pg/mL) in HaCaT cells after (**M**) 24 h and (**N**) 48 h of treatment. Data are presented as means ± SEM of three independent experiments (*n* = 3). Statistical analysis was performed using a one-way ANOVA test followed by a Bonferroni post hoc test (******
*p* < 0.01; *******
*p* < 0.001; ********
*p* < 0.0001 vs. CTRL; **#**
*p* < 0.05; **##**
*p* < 0.01; **###**
*p* < 0.001; **####**
*p* < 0.0001 vs. LPS).

**Figure 3 ijms-24-02721-f003:**
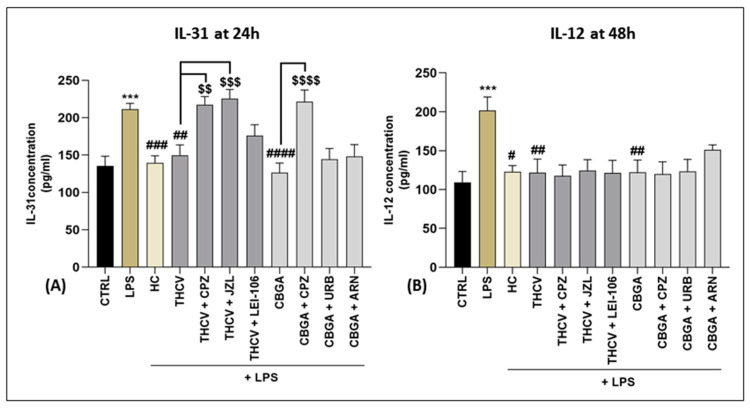
Release of IL-31 at 24 h and IL-12 at 48 h into HaCaT cells medium. (**A**) IL-31 and (**B**) IL-12 expression (pg/mL) in HaCaT cells following 24 h and 48 h treatment with LPS at 5 µg/mL in the presence of HC at 10 µM, including each pCB (THCV at 9.3 µM and CBGA at 13.0 µM) or selected eCB system antagonists and inhibitors (CPZ, 5.0 μM; JZL184, 10.0 μM; LEI-106, 10.0 μM; URB597, 1.0 μM; ARN19874, 33.7 μM). Data are presented as means ± SEM of three independent experiments (*n* = 3). Statistical analysis was performed using a one-way ANOVA test followed by a Bonferroni post hoc test. (*******
*p* < 0.001 vs. CTRL; # *p* < 0.05, **##**
*p* < 0.01, **###**
*p* < 0.001, **####**
*p* < 0.0001 vs. LPS; $$ *p* < 0.01, $$$ *p* < 0.001, $$$$ *p* < 0.0001 vs. the respective pCBs: THCV and CBGA).

**Figure 4 ijms-24-02721-f004:**
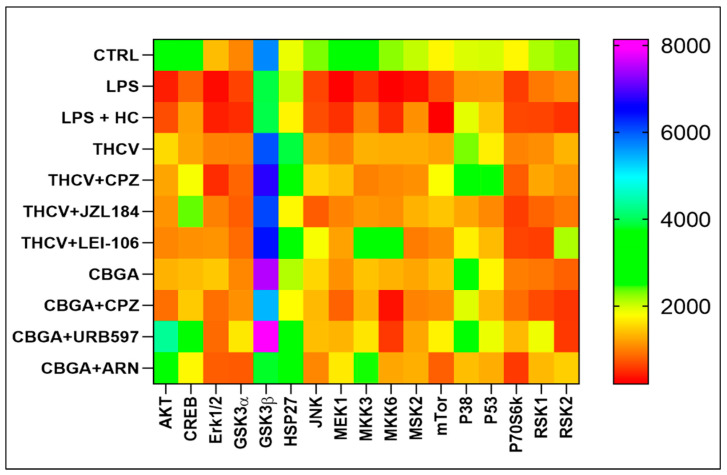
Heat map of the most relevant 17 proteins of the MAPK signaling pathway in HaCaT cells following 24 h treatment with LPS (5.0 µg/mL) in the presence of pCBs (THCV (9.3 µM) and CBGA (13.0 µM)) and selected eCB system antagonists and inhibitors (CPZ (5.0 μM), JZL184 (10.0 μM), LEI-106 (10.0 μM), URB597 (1.0 μM) and ARN19874 (33.7 μM)). Samples were analyzed through the Human Phosphorylation Array, and data are expressed as means of three sets of experiments for each condition (*n* = 3).

**Figure 5 ijms-24-02721-f005:**
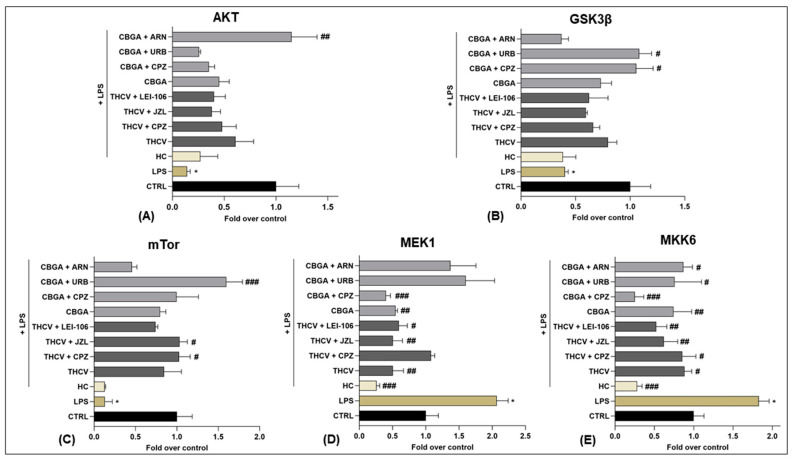
Expression of five of the phosphorylated proteins of the MAPK signaling pathway in HaCaT cells following 24 h treatment with LPS (5.0 µg/mL) in the presence of pCBs (THCV (9.3 µM) and CBGA (13.0 µM)) and selected eCB system antagonists and inhibitors (CPZ (5.0 μM), JZL184 (10.0 μM), LEI-106 (10.0 μM), URB597 (1.0 μM) and ARN19874 (33.7 μM)). Histograms for (**A**) AKT, (**B**) GSK3β, (**C**) mTor, (**D**) MEK1 and (**E**) MKK6. Data are presented as means ± SEM of three independent experiments (*n* = 3). Statistical analysis was performed using a one-way ANOVA test followed by a Bonferroni post hoc test. (* *p* < 0.05 vs. CTRL; # *p* < 0.05, ## *p* < 0.01, ### *p* < 0.001 vs. LPS).

**Figure 6 ijms-24-02721-f006:**
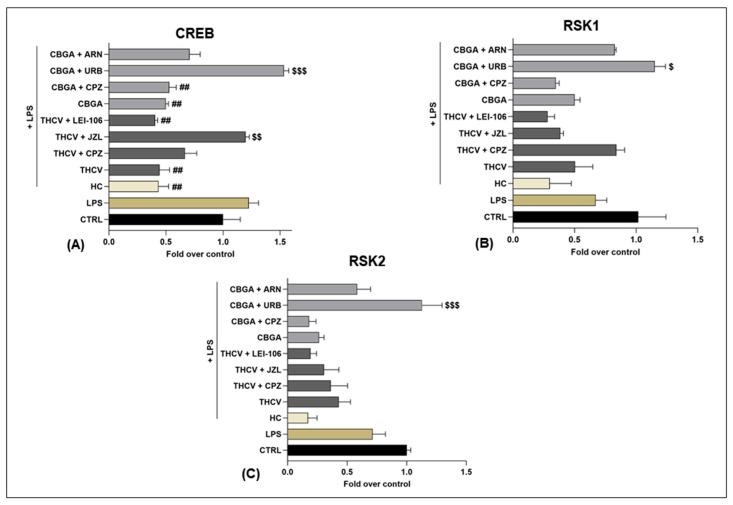
Expression of three of the phosphorylated proteins of the MAPK signaling pathway in HaCaT cells following 24 h treatment with LPS (5.0 µg/mL) in the presence of pCBs (THCV (9.3 µM) and CBGA (13.0 µM)) and selected eCB system antagonists and inhibitors (CPZ (5.0 μM), JZL184 (10.0 μM), LEI-106 (10.0 μM), URB597 (1.0 μM) and ARN19874 (33.7 μM)). Histograms for (**A**) CREB, (**B**) RSK1 and (**C**) RSK2. Data are presented as means ± SEM of three independent experiments (*n* = 3). Statistical analysis was performed using a one-way ANOVA test followed by a Bonferroni post hoc test. (## *p* < 0.01 vs. LPS; $ *p* < 0.05, $$ *p* < 0.01, $$$ *p* < 0.001 vs. the respective pCBs: THCV and CBGA).

**Table 1 ijms-24-02721-t001:** Summary of the most significant effects of pCBs on cytokine release after HaCaT cell treatment with LPS for 24 h and 48 h. Legend: = no significant effect; ↑  significant increase; ↓  significant decrease; + *p* < 0.05; ++ *p* < 0.01; +++ *p* < 0.001; ++++ *p* < 0.0001 vs. LPS.

	CBG		CBC		THCV		CBGA	
*Interleukins*	*24 h*	*48 h*	*24 h*	*48 h*	*24 h*	*48 h*	*24 h*	*48 h*
** *IL-1* ** ** *β* **		↑ ++		↑ +			↓ ++	
** *IL-8* **	↓ ++++		↓ ++		↓ ++		↓ ++	
** *IL-12* **		↓ +++		↓ +++		↓ ++++		↓ ++++
** *IL-31* **					↓ +		↓ ++	

**Table 2 ijms-24-02721-t002:** Summary of the most significant effects of MAPK protein expression in response to eCB system elements in presence of antagonists and inhibitors after HaCaT cells were treated for 24 h with LPS and with THCV and CBGA. Legend: ↑  significant increase; ↓  significant decrease; + *p* < 0.05; ++ *p* < 0.01; +++ *p* < 0.001 vs. LPS, THCV and CBGA.

	THCV Effects				CBGA Effects			
MAPKs	THCV	THCV + TRPV1 Antagonist (CPZ)	THCV + MAGL Inhibitor (JZL184)	THCV + DAGLs Inhibitor (LEI-106)	CBGA	CBGA + TRPV1 Antagonist (CPZ)	CBGA + FAAH Inhibitor (URB597)	CBGA + NAPE-PLD Inhibitor (ARN19874)
**AKT**								↑ ++ vs. LPS
**GSK3** **β**						↑ + vs. LPS	↑ + vs. LPS	
**mTor**		↑ + vs. LPS	↑ + vs. LPS				↑ +++ vs. LPS	
**MEK1**	↓ ++ vs. LPS		↓ ++ vs. LPS	↓ + vs. LPS	↓ ++ vs. LPS	↓ +++ vs. LPS		
**MKK6**	↓ + vs. LPS	↓ + vs. LPS	↓ ++ vs. LPS	↓ ++ vs. LPS	↓ ++ vs. LPS	↓ +++ vs. LPS	↓ + vs. LPS	↓ + vs. LPS
**CREB**	↓ ++ vs. LPS		↑ ++ vs. THCV	↓ ++ vs. LPS	↓ ++ vs. LPS	↓ ++ vs. LPS	↑ +++ vs. CBGA	
**RSK1**							↑ + vs. CBGA	
**RSK2**							↑ +++ vs. CBGA	

**Table 3 ijms-24-02721-t003:** List of primer sequences used for RT-qPCR analysis.

Gene	Forward Primer Sequence (5′→3′)	Reverse Primer Sequence (5′→3′)
PTGS2	ATACTTACCCACTTCAAGGG	ATCAGGAAGCTGCTTTTTACC
ACTB	TGACCCAGATCATGTTTGAG	TTAATGTCACGCACGATTTCC
GAPDH	CAGCCTCAAGATCATCAGCA	TGTGGTCATGAGTCCTTCCA

## Data Availability

Not applicable.

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
