# Peer review of "Rare Phytocannabinoids Exert Anti-Inflammatory Effects on Human Keratinocytes via the Endocannabinoid System and MAPK Signaling Pathway"

_ijms, 2023, doi:10.3390/ijms24032721_

Round 1

Reviewer 1 Report

This study explains antiinflammatory effects after treatment of HaCaT cells with cannabigerol, cannabichromene, Δ9-tetrahydrocannabivarin and cannabigerolic acid, with or without selected inhibitors along MAPK pathway. It is basically clear, objectively proofed and scientifically sound but should have been more beneficial if combined phytocannabinoids were elaborated to indicate their synergistic effects, if any. 

To improve the manuscript, there are some minor revision, as follows:

1. Please check that all materials have already provided companies (with countries), including HC and etc.

2. Please briefly describe and discuss the effect of time on ILs.  

Author Response

Reviewer 1:

This study explains anti-inflammatory effects after treatment of HaCaT cells with cannabigerol, cannabichromene, Δ9-tetrahydrocannabivarin and cannabigerolic acid, with or without selected inhibitors along MAPK pathway. It is basically clear, objectively proofed and scientifically sound but should have been more beneficial if combined phytocannabinoids were elaborated to indicate their synergistic effects, if any. 

Response. We agree that it would be interesting to explore combined phytocannabinoids as well to indicate their potential synergistic effects, but of course this can be done in an independent study. Here, we performed preliminary studies to verify which “minor” phytocannabinoid may have anti-inflammatory effects on keratinocytes. We hope the Referee can accept our point.

To improve the manuscript, there are some minor revisions, as follows:

  1. Please check that all materials have already provided companies (with countries), including HC and etc.
  2. Response: Thank you for your suggestion. We have incorporated this information in the amended manuscript.
  3. Please briefly describe and discuss the effect of time on ILs. 
  4. Response: We added a short comment on the time-dependent release of interleukins (IL-8, IL-12, IL-31) in the Discussion (page 10, lines 300-305), as suggested.

Reviewer 2 Report

Thank you for an interesting manuscript. I enjoyed reading it!

My foremost concern is the material and methods section, I lack the method for OD measurement and analysis, some parts of the RT qPCR method and some general details about sample storage and centrifugation.

In particularly, I lack the description of the OD measurements. I find this methodology to be used in several figures and to be central for your conclusions? Unless I have misinterpreted the methodology, in which case I hope you can make it more clear how the data was obtained.

Comments:
Line 21, 28, 88, 95, 352: I have never seen the term interrogate be used instead of
investigate. Are you sure of this?

Line 42-44: In the last few years, the use of botanical ingredients in dermatology has been
continuously increasing, as they are considered safer and often more effective than other
pharmaceutical drugs [1].

I question the consensus that botanical ingredients are considered more effective and safer.
It is not the conclusion by the cited review. I suggest a more modest statement: “In the last
few years, the use of botanical ingredients in dermatology has been continuously
increasing,”... studies have found different botanical substances both effective and safe
although more studies and clinical trials are needed [1].

“ A plant that has a long history of medical as wells as recreational purposes is cannabis.
Recently, the interest in the medical aspects have increased and the legalization of cannabis
in a growing number of countries add to the possibility to use cannabinoids in healthcare
and skincare formulation [2].

Line 60-73/74-87: The description is a general description of the endocannabinoid system.
Can you be more specific about the eCB system in skin? How is the distribution of CB
receptors in the skin? What is known about the distribution of the CB receptors and
enzymes in skin? Perhaps flip the sections presenting the description of the skin (line 74-87)
first followed by the description of the eCB system (Line 60-73) and its role in the skin. Self
citation is something we should avoid but I read your article, reference [8] and it gave me a
lot of the background I wished was in this paper. This manuscript is a logical follow up on
your paper [8]. Could you, in this paper, describe your findings and how it lead you to the
experiments and hypothesis investigated in.

Line 139: do you mean incidentally as in “by the way” or “as a chance occurance” or initially
as in “at first”?

Line 139: the generation of the dose respons curve (better called standard curves?) is not
included in the material and methods section. It is also referred to as preliminary. What is
preliminary about the experiment and its results? You have n=3 as in your other experiments
and you present the data in its full? This method is used for figure 2 and 3 so the method is
very important for the evaluation of your results. What OD wavelength was used; what
medium was used? Was the OD measured in the same medium as the cells was later in?
How was the samples collected? How do you identify and separate the different cytokines?

Could there be confounding factors in the cell supernatants that make your assumptions and
extrapolations invalid?

Line 144 and 161, 164, 176, 177: change “the expression of” to “the release of” since that is
what you are measuring.

Figure 2: consider increasing Y-axis legend.

Table 2: clarify what alone means in the table, untreated, control? If all are treated with
LPS then LPS could be stated in the “alone” box.

Line 396: at what speed did you centrifuge the cell pellets and supernatants?

Line 400: Be more precise, the RevertAid H Minus First Strand cDNA Synthesis Kit

was used to synthesize cDNA of isolated RNA. How (what kit) was used for RNA isolation?
Was the isolation done immediately after the experiment or was the cell pellets stored?

Line 402: The description of the qPCR needs more information. Consult the MIQE guidelines
for guidance on information that is considered important to declare. I lack information about
the complete reaction conditions (cycles, temperatures, amount of cDNA), the efficiency of
the primers is also useful information to validate the results. Note that you haven’t used the
same abbreviation in line 106 and the rest of the manuscript. You have done a real time
quantitative reverse transcription PCR RT qPCR (abb. acc. to MIQE guidelines).

(Bustin SA, Benes V, Garson JA et al., The MIQE Guidelines: Minimum Information for
Publication of Quantitative Real-Time PCR Experiments, Clinical chemistry (2009)55:4;611-
622.).

Line 425: one assumes that the concentrations are specific and pre-calculated. Maybe a
better description is “concentration known to do something.... [8]”

Line 431: centrifuged for how long and at what speed? Check that all centrifugation steps in
the manuscript is given with the time and speed.

Line 466: you have nicely declared the number on n (=3) in each experiment but how about
the number of N ́s?

Line 470-473: this conclusion does not match the purpose you present in Line 88-90.

Line 321: I don’t follow the conclusion. Do you suggest a link between 2Ag and IL-31? Is
there any support for this conclusion?

It would be interesting to see the effects of CB1 and CB2 antagonists in your model. In line
178-179 you write “modulation of eCB system elements like metabolic enzymes and/or
binding receptors”. The same is seen in line 212, you write receptors but only include a
TRPV1 antagonist?

Line 342: In figure 3: URB397 does not revert the decrease in IL-31 mediated by CBGA.

Supplementary Fig.2: The circled pair (Akt) in the top left of the CTRL slide has not been
named. Table 4 does not add much, I suggest presenting it with supplementary Fig.2. in
supplementary info.

Thank you and good luck!

Kind regards

Author Response

Reviewer 2:

Thank you for an interesting manuscript. I enjoyed reading it!

My foremost concern is the material and methods section, I lack the method for OD measurement and analysis, some parts of the RT qPCR method and some general details about sample storage and centrifugation.

In particularly, I lack the description of the OD measurements. I find this methodology to be used in several figures and to be central for your conclusions? Unless I have misinterpreted the methodology, in which case I hope you can make it more clear how the data was obtained.

Response: Thanks for pointing this out. We agree with this comment. Therefore, we have improved the Materials and Methods section in the revised manuscript.

Comments:

  1. Line 21, 28, 88, 95, 352: I have never seen the term interrogate be used instead of investigate. Are you sure of this?

1.Response: Thanks for the tip, we have edited as suggested (lines 21, 28, 104, 353).

  1. Line 42-44: In the last few years, the use of botanical ingredients in dermatology has been continuously increasing, as they are considered safer and often more effective than other pharmaceutical drugs [1]. I question the consensus that botanical ingredients are considered more effective and safer. It is not the conclusion by the cited review. I suggest a more modest statement: “In the last few years, the use of botanical ingredients in dermatology has been continuously increasing,”... studies have found different botanical substances both effective and safe although more studies and clinical trials are needed [1]. “ A plant that has a long history of medical as wells as recreational purposes is cannabis. Recently, the interest in the medical aspects have increased and the legalization of cannabis in a growing number of countries add to the possibility to use cannabinoids in healthcare and skincare formulation [2].
  2. Response: Thank you for your kind suggestions. We have revised the sentence, as recommended (pages 1 and 2, lines 43-48).

  3. Line 60-73/74-87: The description is a general description of the endocannabinoid system. Can you be more specific about the eCB system in skin? How is the distribution of CB receptors in the skin? What is known about the distribution of the CB receptors and enzymes in skin? Perhaps flip the sections presenting the description of the skin (line 74-87) first followed by the description of the eCB system (Line 60-73) and its role in the skin. Self citation is something we should avoid but I read your article, reference [8] and it gave me a lot of the background I wished was in this paper. This manuscript is a logical follow up on your paper [8]. Could you, in this paper, describe your findings and how it lead you to the experiments and hypothesis investigated in.
  4. Response: We have included in the Introduction a sentence in order to briefly explain the distribution of ECS in different skin cells (page 2, lines 75-79),as well as a new short sentence (page 3, lines 98-102) on the effects of phytocannabinoids in relation to ECS in HaCaT cells, as reported in our previous article. We hope that Referee can accept these points.

  1. Line 139: do you mean incidentally as in “by the way” or “as a chance occurance” or initially as in “at first”? Line 139: the generation of the dose-response curve (better called standard curves?) is not included in the material and methods section. It is also referred to as preliminary. What is preliminary about the experiment and its results? You have n=3 as in your other experiments and you present the data in its full? This method is used for figure 2 and 3 so the method is very important for the evaluation of your results. What OD wavelength was used; what medium was used? Was the OD measured in the same medium as the cells was later in? How was the samples collected? How do you identify and separate the different cytokines? Could there be confounding factors in the cell supernatants that make your assumptions and extrapolations invalid?
  2. Response: Thank you for your kind suggestion. We accept your point of view, and thus we have improved the information related to ELISA experiments, in order to clarify them as much as possible (page 4, lines 146-149). In addition, we replaced “dose-response” with “standard” curves (line 148) and we have added more information on ELISA assay protocol in the paragraph 4.4 of Materials and Methods (page 13, lines 442-446). Please note that OD values are reported in lines 452-455 (page 13). As for the number of samples and experiments, we performed each analysis on three different batches of HaCaT cells (N=3) and each biological sample was replicated three times.

  1. Line 144 and 161, 164, 176, 177: change “the expression of” to “the release of” since that is what you are measuring. Figure 2: consider increasing Y-axis legend. Table 2: clarify what “alone” means in the table, untreated, control? If all are treated with LPS then LPS could be stated in the “alone” box.
  2. Response: We have changed “the expression of” with “the release of” and modified the Y-axis, as suggested. We have changed the “alone” with the name of specific pCB treatment in the Table 2 of the revised version.

  1. Line 396: at what speed did you centrifuge the cell pellets and supernatants?
  2. Response: We have added this information in the Materials and Methods section, as suggested (page 12, line 398).

  1. Line 400: Be more precise, the RevertAid H Minus First Strand cDNA Synthesis Kit was used to synthesize cDNA of isolated RNA. How (what kit) was used for RNA isolation? Was the isolation done immediately after the experiment or was the cell pellets stored?
  2. Response: Thank you for your kind suggestions. More information has included in the revised version (Paragraph 4.3 of Materials and Methods).

  1. Line 402: The description of the qPCR needs more information. Consult the MIQE guidelines for guidance on information that is considered important to declare. I lack information about the complete reaction conditions (cycles, temperatures, amount of cDNA), the efficiency of the primers is also useful information to validate the results. Note that you haven’t used the same abbreviation in line 106 and the rest of the manuscript. You have done a real time quantitative reverse transcription PCR – RT qPCR (abb. acc. to MIQE guidelines). (Bustin SA, Benes V, Garson JA et al., The MIQE Guidelines: Minimum Information for Publication of Quantitative Real-Time PCR Experiments, Clinical chemistry (2009)55:4;611- 622.).
  2. Response: Thank you for the suggestions. We agree that the description was incomplete, thus we have now included the missing information in the paragraph 4.3 of the Materials and Methods (page 12).

  1. Line 425: one assumes that the concentrations are specific and pre-calculated. Maybe a better description is “concentration known to do something.... [8]”
  2. Response: Thank you for your kind suggestions. We have included a new sentence (page 13, line 431) in order to specify the meaining of the used pCBs concentrations. We hope the Referee can accept our point.

  1. Line 431: centrifuged for how long and at what speed? Check that all centrifugation steps in the manuscript is given with the time and speed.
  2. Response: We have included this information in the revised manuscript, as suggested (page 13, line 436).

  1. Line 466: you have nicely declared the number on n (=3) in each experiment but how about the number of N ́s?
  2. Response: We performed all experiments on N=3 different batches of HaCaT cells, each replicated 3 times.

  1. Line 470-473: this conclusion does not match the purpose you present in Line 88-90.
  2. Response: We have modified the sentence in the conclusion to better clarify the purpose of our study (page 14, line 481). We hope the Referee can accept this point.

  1. Line 321: I don’t follow the conclusion. Do you suggest a link between 2Ag and IL-31? Is there any support for this conclusion? It would be interesting to see the effects of CB1 and CB2 antagonists in your model.
  2. Response: The effect of THCV on IL-31 is mediated by 2-AG metabolism, because the blockade of 2-AG degradation by JZL184 reverted the THCV-induced reduction on IL-31. Thus, the resulting increased concentration of 2-AG can be considered proinflammatory. This point has been clarified in the Discussion (page 11, lines 323-324).

Thank you for your kind suggestions on CB receptors. It would be interesting to test the effect of CB1 and CB2 antagonists in our model, certainly we hope to be able to follow this suggestion in future experiments.

  1. In line 178-179 you write “modulation of eCB system elements like metabolic enzymes and/or binding receptors”. The same is seen in line 212, you write receptors but only include a TRPV1 antagonist?
  2. Response: We have modified the sentence by changed “binding receptors” with “TRPV1 receptor” in the revised version (page 6, line 184; page 7, line 217).

  1. Line 342: In figure 3: URB397 does not revert the decrease in IL-31 mediated by CBGA.
  2. Response: Thank you for underlining the mistake, that we amended (page 11, line 344) in the revised manuscript.

  1. Supplementary Fig.2: The circled pair (Akt) in the top left of the CTRL slide has not been named.
  2. Response: We have included the names of Akt and GSK3β in the Supplementary Fig. 2, as suggested.

  1. Table 4 does not add much, I suggest presenting it with supplementary Fig.2. in supplementary info.
  2. Response: We agree and have moved the Table 4 to Supplementary Materials, as suggested. We renamed this table as Table S1.